# Aberrant DNA Methylation in Acute Myeloid Leukemia and Its Clinical Implications

**DOI:** 10.3390/ijms20184576

**Published:** 2019-09-16

**Authors:** Xianwen Yang, Molly Pui Man Wong, Ray Kit Ng

**Affiliations:** 1School of Biomedical Sciences, Li Ka Shing Faculty of Medicine, The University of Hong Kong, Hong Kong SAR, China; 2School of Biomedical Sciences, Faculty of Medicine, The Chinese University of Hong Kong, Hong Kong SAR, China; 3The University of Hong Kong Shenzhen Institute of Research and Innovation, The University of Hong Kong, Hong Kong SAR, China

**Keywords:** Acute myeloid leukemia, DNA methylation, epigenetic biomarker, therapeutics

## Abstract

Acute myeloid leukemia (AML) is a heterogeneous disease that is characterized by distinct cytogenetic or genetic abnormalities. Recent discoveries in cancer epigenetics demonstrated a critical role of epigenetic dysregulation in AML pathogenesis. Unlike genetic alterations, the reversible nature of epigenetic modifications is therapeutically attractive in cancer therapy. DNA methylation is an epigenetic modification that regulates gene expression and plays a pivotal role in mammalian development including hematopoiesis. DNA methyltransferases (DNMTs) and Ten-eleven-translocation (TET) dioxygenases are responsible for the dynamics of DNA methylation. Genetic alterations of DNMTs or TETs disrupt normal hematopoiesis and subsequently result in hematological malignancies. Emerging evidence reveals that the dysregulation of DNA methylation is a key event for AML initiation and progression. Importantly, aberrant DNA methylation is regarded as a hallmark of AML, which is heralded as a powerful epigenetic marker in early diagnosis, prognostic prediction, and therapeutic decision-making. In this review, we summarize the current knowledge of DNA methylation in normal hematopoiesis and AML pathogenesis. We also discuss the clinical implications of DNA methylation and the current therapeutic strategies of targeting DNA methylation in AML therapy.

## 1. Introduction

Definitive hematopoiesis refers to the hierarchical process in which hematopoietic stem cells (HSCs) undergo differentiation and give rise to all various lineages of mature hematopoietic cells. HSCs, which reside at the apex of the hierarchy, possess the ability of self-renewal and multi-lineage differentiation, whereas progenitors have a limited capacity of self-renewal and are committed to restricted hematopoietic lineages. Hematopoiesis is a complicated process including HSC specification, expansion, and differentiation. It requires precise orchestration of the transcriptional regulation and signal transduction within the cells and with the bone marrow niche [1,2,3]. In the past decade, mounting evidence suggested that epigenetic mechanisms, such as DNA methylation, that regulate transcriptional program are crucial in hematopoiesis [4,5].

In the mammalian genome, DNA methylation denotes the addition of methyl group to the fifth carbon position of cytosine, which is frequently occurring at CpG dinucleotides. It is involved in various cellular processes, particularly in genomic imprinting, X-chromosome inactivation, repression of transposable elements, and regulation of gene expression [6,7]. DNA methylation is catalyzed by three conserved DNA methyltransferases (DNMTs), which include the de novo DNMT3A and DNMT3B, and the maintenance DNMT1. DNMT3A and DNMT3B are responsible for the establishment of DNA methylation at the unmodified DNA template, whereas DNMT1 can faithfully propagate the pre-existed methylation pattern in the parental DNA strand to the newly synthesized daughter strand during DNA replication. On the other hand, the removal of DNA methylation is mediated by the Ten-Eleven-Translocation (TET) dioxygenase family [8]. TET family proteins are able to convert 5-methylcytosine (5mC) to 5-hydroxymethylcytosine (5hmC) and other derivatives, e.g., 5-formylcytosine (5fC) and 5-carboxylcytosine (5caC), which can be excised from the DNA strand through the base excision repair mechanism and subsequently replaced by an unmodified cytosine [9,10,11].

Over the past decades, the advancement of technology allows a comprehensive analysis of genome-wide DNA methylation patterns in hematopoietic lineage cells and demonstrates that DNA methylome undergoes extensive changes during hematopoietic development [12,13,14]. Importantly, the dynamics of DNA methylation is not a consequence of hematopoietic differentiation, but rather plays a causative role in cell fate determination, which is illustrated by animal models with the disruption of *DNMT* genes [15,16]. It has been reported that the development of hematological malignancies, such as acute myeloid leukemia (AML), is closely associated with an altered DNA methylation and the concomitant transcriptional dysregulation [4,5,17]. AML is a fatal disease, which is characterized by clonal expansion of leukemic blast in bone marrow, peripheral blood, and other organs. Treatment of AML remains a major challenge owing to the diverse molecular mechanisms associated with the oncogenic events [18,19]. Interestingly, aberrant DNA methylation was reported as one of the hallmarks of AML, which could be used for the stratification of AML patients and the identification of methylated gene sets as biomarkers for therapeutic decision and outcome prediction [20,21,22]. In this review, we focus on the current knowledge of DNA methylation in AML and evaluate its clinical implications in AML diagnosis and prognosis.

## 2. DNA Methylation Enzymes in Hematopoiesis

DNA methylation plays an important role in mammalian development. The profound effect of DNA methylation in hematopoiesis has been demonstrated by numerous studies using knockout mouse models of the DNA methylation enzymes (Figure 1). The maintenance DNMT1 was shown to be required for HSC self-renewal, homing and suppression of apoptosis [15,23]. Conditional knockout of *Dnmt1* in the *Mx1*-Cre mice resulted in a reduction of HSC self-renewal and niche retention, which was partially caused by the downregulation of *Cd62l* and *Ski* genes [23]. Besides, DNMT1 is important in the regulation of the myeloid/lymphoid lineage commitment. DNMT1-deficient HSCs showed biased differentiation to the myeloid lineage, which was associated with the reactivation of myeloerythroid-specific genes, such as *Cd48* and *Gata1*, through reduction of promoter DNA methylation [15]. This is also supported by the findings from several studies which showed that the myeloid-specific loci were hypermethylated in lymphoid progenitors [13,24,25].

The function of de novo DNMT3A and DNMT3B in hematopoiesis was first reported by conditional knockout HSC models. Single knockout of *Dnmt3a* or *Dnmt3b* in HSCs showed no significant impact on self-renewal or differentiation; in contrast, only the double knockout HSCs failed in long-term reconstitution of myeloid and B lymphoid lineages over serial transplantation [16]. However, *DNMT3A* mutations showed high prevalence in leukemia patients, with 20% in AML and 10% in myelodysplastic syndrome (MDS) [26,27], which led to further investigation of the role of DNMT3A in hematopoiesis. Challen et al. reported that *Dnmt3a*^-/-^ HSCs repopulated extensively in the recipient mice and generated at least 15 times more in HSC numbers when compared to the recipients of normal HSCs. The *Dnmt3a*^-/-^ HSCs also exhibited a higher in vitro colony-forming activity. However, the in vivo differentiation potential of *Dnmt3a*^-/-^ HSCs sharply declined. The phenotypes of *Dnmt3a*^-/-^ HSCs were associated with the aberrant DNA methylation pattern which dysregulated the expression of genes involved in differentiation (e.g., *Flk2*, *Pu.1*, and *Ikaros*) and multipotency (e.g., *Runx1*, *Gata3*, *Pbx1*, *Foxo1* and *Cdkn1a*) [28]. The null phenotype of *Dnmt3* knockout HSCs in the earlier report might be caused by the low efficiency of gene targeting with the retroviral Cre expression system or the use of less stringent approaches in assessing the HSC properties. Moreover, a recent study showed that the self-renewal capacity of HSCs with *Dnmt3a* deletion can be maintained for at least 12 transplant generations, which is far more than that of transplanting normal HSCs. It was further revealed that serial transplantation of *Dnmt3a*^-/-^ HSCs resulted in a gradual loss of DNA methylation at the key HSC regulatory elements, thereby stabilizing the epigenetic features associated with self-renewal [29]. On the other hand, the function of DNMT3B is less profound in hematopoiesis. Deletion of *Dnmt3b* showed only a mild in vivo HSC phenotype, which can be compensated by *DNMT3A*. Although DNMT3B shows less potency in HSCs, analysis of the mutant methylome revealed that CpG island hypermethylation can be triggered by the abnormal activity of DNMT3B when DNMT3A was absence [30]. As a result, DNMT3B seems to have a redundant, yet synergistic, role with DNMT3A in HSCs.

TET protein family is another important group of epigenetic regulators involved in hematopoietic development. *TET1* is mainly expressed in embryonic stem cells, while *TET2* and *TET3* are expressing in a vast majority of adult tissues. Deletion of *Tet1* in mice showed mild lymphocytosis with the enrichment of mature B cells in the peripheral blood, lymph nodes and spleen [31]. Transplantation of *Tet1*^-/-^ HSCs demonstrated an enhanced reconstitution frequency with a biased differentiation towards the B cell lineage in the recipient mice, suggesting that TET1 is crucial in lymphoid differentiation. Despite the fact that constitutive deletion of *Tet3* in mice is embryonic lethal, conditional knockout in the hematopoietic lineage showed a minor increase in the number of HSCs but without perturbing the frequencies of differentiated hematopoietic cells [32]. Among the three TET family members, *TET2* shows a strong expression in hematopoietic lineage and undergoes somatic mutations frequently in hematological malignancies. It has been reported that loss of *Tet2* in murine bone marrow cells dramatically reduced the global 5hmC level and concomitantly increased the 5mC level [33]. *Tet2* knockout mice showed defects in bone marrow with enlargement of the HSC compartment, and eventually developed splenomegaly, monocytosis and extramedullary hematopoiesis [34]. HSCs with *Tet2* deletion displayed enhanced self-renewal capacity, which allowed them to markedly outcompete the wild-type counterparts and predominate in the peripheral blood of the transplanted mice [33]. Moreover, *Tet2*^-/-^ HSCs displayed a transcriptional program resembling that of the common myeloid progenitors, but with an increased expression of self-renewal regulators *Meis1* and *Evi1*, and a reduced expression of myeloid-specific factors *Cebpa*, *Mpo*, and *Csf1* [33]. These results indicated that TET2 is essential in HSC self-renewal and differentiation towards the myeloid lineage [33,34]. In addition, a recent study revealed a novel function of TET2 in the protection of genomic stability [35]. A higher frequency of chromosomal alterations was observed in hematopoietic stem and progenitor cells (HSPCs) with *Tet2* deletion, which affected the genomic loci containing genes that are associated with hematological malignancies, e.g., *Flt3*, *Notch1*, and *Mll2*. Interestingly, a majority of these mutated sites were featured by an accumulation of 5hmC, which could not be further oxidized to 5fC and 5caC due to the absence of TET2 [35]. Therefore, TET2 can safeguard the genome mutagenicity and function as a tumor suppressor in hematopoiesis. Although depletion of any TET proteins resulted in varying aspects of hematopoietic dysregulation, *Tet1* and *Tet2* double knockout mouse model exhibited higher incident of B-cell malignancies when comparing to the single knockout mice [36]. Combined deletion of *Tet2* and *Tet3* also promoted a more rapid development of myeloid malignancies [37]. These studies thus suggest the redundant functions of TET proteins in normal hematopoietic development.

## 3. Aberrant DNA Methylation in AML

Alterations in DNA methylation landscape are frequently observed in multiple types of cancers. In general, cancer cells present a global hypomethylated genome with specific CpG island hypermethylation, which collectively leads to genomic instability and aberrant gene expression. It has been reported that dysregulation of DNA methylation is associated with hematological malignancies. Different subtypes of AML were reported having distinct DNA methylation profiles [18]. However, it remains controversial whether the altered DNA methylation is a consequence or a cause of AML. In this section, we discuss the role of DNA methylation in the pathogenesis of AML with various genetic abnormalities generated from chromosomal translocations or single gene mutations.

### 3.1. Acute Promyelocytic Leukemia (APL) with PML-RARα

APL is characterized by the presence of *PML-RAR**α* fusion gene which is resulted from t(15;17) chromosomal translocation. The fusion protein PML-RARα is responsible for differentiation blockage at the promyelocytic stage through inactivation of genes associated with cell differentiation and apoptosis [38,39]. It was reported that PML-RARα requires DNMT3A to act as an oncogenic transcription factor in APL initiation. The DNA methyltransferase activity of DNMT3A was demonstrated to be indispensable to the enhanced self-renewal of PML-RARα-transformed hematopoietic progenitors [40]. Mechanistically, two studies from the same group have reported that PML-RARα recruited different repressive epigenetic modifiers, such as DNMT3A [41] or MBD1 [42], to mediate DNA hypermethylation at the *RAR**β2* promoter for gene silencing. However, these studies only focused on a limited number of loci, such that the global association between PML-RARα and DNA methylation in leukemogenesis remains unclear. With the technological advancement of genome sequencing, several studies demonstrated a limited role of DNA methylation in APL pathogenesis [43,44]. By utilizing reduced representation bisulfite sequencing (RRBS) analysis, Schoofs et al. observed no apparent differences in the global DNA methylation pattern by comparing bone marrow cells from pre-leukemic *PML-RAR**α* knock-in mice with the normal counterparts [43]. A majority of PML-RARα binding sites (547 out of 556 binding sites) failed to overlap with the APL patient-specific differentially methylated regions, whereas only 3 PML-RARα binding sites, including *RAR**β2* promoter, were hypermethylated. These findings suggest that PML-RARα binding sites are not epigenetically inactivated by the DNA methylation machinery. This is indeed supported by other genome-wide studies showing that PML-RARα induced a hypo-acetylated chromatin state through the recruitment of histone deacetylase in APL cells [44,45]. Interestingly, treatment of all-*trans* retinoic acid to APL patients can efficiently reverse the repressive function of PML-RARα on the differentiation transcriptional program without significant alterations in the DNA methylation pattern, suggesting that DNA methylation may not be crucial in the maintenance of APL [43,44]. Even though the recruitment of DNMT3A by PML-RARα is an important event for APL initiation, the establishment of APL-specific DNA methylation signature is likely a secondary event during the course of APL progression.

### 3.2. AML with MLL Gene Rearrangement

Chromosomal translocations involving *Mixed Lineage Leukemia* (*MLL*) gene at chromosome 11q23 are reported in 7% of adult AML patients. Up till now, more than 70 partner genes are found to be fused with the *MLL* gene, resulting in the generation of MLL fusion proteins. The diversity of fusion partners indeed complicates the pathogenesis of *MLL*-rearranged AML. Genome-wide analysis of DNA methylation by *Hpa*II Tiny Fragment Enrichment by Ligation-Mediated PCR (HELP) assay showed a unique methylation profile with significant hypomethylation in the AML patients with 11q23 abnormalities [18]. Notably, overexpression of *MLL-AF9* in human HSPCs resulted in a distinct DNA methylation signature which resembles that of the MLL-AF9 AML patients [46], suggesting a close correlation between MLL fusion protein and aberrant DNA methylation in leukemic transformation. We also observed that the *Hoxa* promoters were aberrantly hypomethylated in a MLL-EEN leukemia mouse model, which accounts for the upregulated expression of *Hoxa* cluster genes [47]. Importantly, the CXXC domain retained in the MLL fusion proteins can recognize non-methylated CpG dinucleotides at *Hoxa9* locus and protect it from DNA methylation and repressive histone 3 lysine 9 tri-methylation (H3K9me3) mark, which subsequently triggers an aberrant induction of *Hoxa9* [48,49]. Interestingly, a recent study demonstrated that DNMT3A is dispensable for the oncogenic property of MLL-AF9. Ectopic expression of MLL-AF9 in murine bone marrow cells induced self-renewal ex vivo, and produced a rapid-onset and high-penetrance AML phenotype in vivo regardless of the cellular DNMT3A status [40]. Indeed, deletion of *Dnmt1* prevents the development of MLL-AF9 leukemia in vivo [50], suggesting that MLL fusion proteins require maintenance methylation, but not de novo methylation, for leukemia induction.

### 3.3. AML with DNMT3A Mutations

Around 25% of AML patients harbor *DNMT3A* mutations [17]. Interestingly, the majority of *DNMT3A* mutations in AML are heterozygous, whereas homozygous mutations predispose the development of T lymphoid leukemia [17]. HSCs bearing *DNMT3A* mutations are in a pre-leukemic state [51,52], which readily undergo malignant transformation by acquiring additional genetic alterations, e.g., mutations in *FLT3*, *NPM1*, and *IDH1* [26]. Although the de novo methyltransferases DNMT3A and DNMT3B show redundant functions, it was found that DNMT3B was barely detectable in AML cells [53]. It thus suggests that the de novo methylation activity is mainly contributed by DNMT3A in AML. Importantly, 60% of *DNMT3A* mutations in AML patients occur at the residue R882, which is located at the methyltransferase catalytic domain [26,54,55]. The DNMT3A^R882H^ mutant not only loses its methyltransferase activity but also functions as a dominant negative mutant that reduces the enzymatic activity of the wild-type DNMT3A by over 80% [53]. Focal hypomethylation at specific CpG sites was observed in primary AML patients with *DNMT3A^R882H^* [53,56]. DNMT3A^R882H^ can trigger DNA hypomethylation and enhance the binding of active histone modifiers at the enhancer elements, resulting in aberrant transactivation of leukemic stemness genes, such as *Mn1*, *Hoxa* gene cluster, and HOX cofactor *Meis1* [57]. Other studies also reported that DNA hypomethylation mediated by DNMT3A^R882H^ is an initiating event in AML development [51,58,59,60,61]. Importantly, data from The Cancer Genome Atlas (TGCA) showed that many AML patients have co-occurrence of *DNMT3A*, *FLT3* and *NPM1* mutations. Knock-in mouse model with the triple gene mutations (*Dnmt3a^mut^*/*Flt3^ITD^*/*Npm1^c^*) displayed a fully penetrant leukemic phenotype, whereas the double mutation models (*Dnmt3a^mut^* with either *Flt3^ITD^* or *Npm1^c^*) exhibited long latency and low penetrance [62]. The triple mutant bone marrow cells also demonstrated enhanced engraftment efficiency than the double mutant cells. A recent study further shows that the triple-mutated AML patient samples contained a higher frequency of leukemic stem cell, which is associated with the hypomethylation of the *Hepatic Leukemia Factor* (*HLF*) gene [63]. These findings thus support that *DNMT3A* mutation can epigenetically cooperate with other genetic alterations in leukemic transformation.

### 3.4. AML with TET2 Mutations

TET2 is another epigenetic modifier which is frequently mutated in hematological malignancies, including chronic myelomonocytic leukemia (CMML), MDS/myeloproliferative neoplasms (MPNs), AML, and T- or B-cell lymphoma [64,65,66,67]. Around 17% of AML patients carry loss-of-function mutations of *TET2* [68]. *Tet2* mutations can prime HSCs to a pre-leukemic state which retains the ability to differentiate into a full spectrum of mature blood cells. However, these pre-leukemic stem cells can become leukemic initiating cells after the acquisition of additional genetic lesions, leading to the development of full-blown leukemia [52,69,70]. This indicates that *TET2* mutations *per se* can promote the leukemic transformation, but they are not sufficient to complete the whole process. Mutational landscape analysis in AML also revealed that *TET2* mutations frequently co-occur with other mutations in *NPM1*, *FLT3*, *JAK2*, *RUNX1*, *CEBPA*, and *KRAS* [71], suggesting that TET2 inactivation cooperates with these additional mutations in driving leukemogenesis. This is further supported by the findings that depletion of *Tet2* together with *Flt3-ITD* mutation can synergistically dysregulate DNA methylation and interfere with normal hematopoietic differentiation, resulting in an accumulation of HSPC and GMP populations [72]. Of note, a majority of the hypermethylated regions under *Tet2* and *Flt3-ITD* mutations are located at gene regulatory elements, resulting in deregulation of genes involved in self-renewal and differentiation (*Id1*, *Gata1*, *Gata2*, *Mpl*, and *Socs2*) [72]. Besides, knockout of *Tet2* in pre-leukemic cells with AML1-ETO resulted in genome-wide DNA hypermethylation, which affects nearly 25% enhancer elements [73]. As the majority of these hypermethylated enhancers are associated with tumor suppressor genes, it thus proposes an epigenetic mechanism that is associated with *TET2* mutations in leukemia development.

Interestingly, it was observed that mutations of *TET2* and *IDH1/2* were found mutually exclusive in AML patients. The DNA methylation signature in AML patients carrying *IDH1/2* mutations was partially overlapped with those carrying *TET2* mutations, suggesting that these two mutations deploy the same DNA methylation pathway in AML pathogenesis [74]. As TET2 mediates DNA demethylation through 5mC hydroxylation in an α-ketoglutarate-dependent manner, 2-hydroxyglutarate generated by the mutant IDH1/2 proteins can serve as a competitive inhibitor of TET2 [75]. As a result, *IDH1/2* mutations promote leukemia development by interfering myeloid differentiation and aberrantly enhancing c-Kit expression through the disruption of TET2-mediated DNA demethylation [74].

## 4. Clinical Implications of DNA Methylation in AML

AML patients can be stratified into three major risk subgroups (favorable, intermediate, and poor outcomes) based on the cytogenetic analysis [76]. The most common cytogenetic abnormalities in the favorable-risk subgroup include t(15;17), t(8;21), and inv(16), whereas AML patients with adverse prognosis have abnormal 3q, 11q23 abnormalities other than t(9;11), and complex karyotypes [77]. The intermediate-risk subgroup mainly comprises of AML patients with normal karyotypes, which is known as cytogenetically normal AML (CN-AML) [78]. Since 50% of adult AML are classified as CN-AML, it represents patients with a very broad diversity of molecular characterization and clinical outcomes [79,80]. Therefore, the existing classification of AML imposes a challenge in prognosis and precision treatment owing to the heterogeneity within each risk subgroup. The advancement of the next generation sequencing technology moves the field of AML biology forward through robust identification of genetic alterations in CN-AML, such as mutations in *NPM1*, *FLT3*, and *CEBPA* genes. Interestingly, it has been demonstrated that AML patients with different cytogenetic or genetic alterations show distinct global patterns of DNA methylation [18,81,82]. It thus implies that DNA methylation may serve as an additional parameter in the stratification of AML patients. At a glance, many studies have already investigated the clinical implications of DNA methylation in AML (Table 1) [18,21,83,84,85,86,87,88,89,90,91,92,93,94]. In this section, we discuss the potential of DNA methylation assessment in AML diagnosis and prognosis, and in the evaluation of therapeutic efficacy.

### 4.1. Diagnostic Value of DNA Methylation

Clonal hematopoiesis, which is a strong risk factor for subsequent hematological malignancies, was reported to be frequently associated with somatic mutations of *DNMT3A* and *TET2* [59]. Since *DNMT3A* and *TET2* mutations can prime HSCs to a pre-leukemic state through dysregulation of DNA methylation [51,52], it is proposed that assessing the aberrant DNA methylation pattern is valuable for an early detection of AML with no clinical manifestations. An early attempt of using meta-analysis for evaluation of the correlation between aberrant DNA methylation and AML risk prediction was conducted by re-analyzing 41 case-control studies. This study identified three candidates, *CDKN2A*, *CDKN2B*, and *ID4*, that were significantly hypermethylated in AML and were associated with the increased risk of developing leukemia [95]. However, most of the case-control studies used methylation-specific PCR for qualitative measurement of the DNA methylation status, which limits the quantitative comparison of the whole spectrum of candidate genes. Another study used bisulfite pyrosequencing to screen for methylated CpG islands in 21 leukemic cell lines and 30 primary patient samples had identified 8 genes (*NOR1*, *CDH13*, *CDKN2B*, *NPM2*, *OLIG2*, *PGR*, *HIN1*, and *SLC26A4*) that were frequently methylated. Strikingly, the methylation levels of many of these candidate genes were found statistically higher in the relapse samples when compared to the samples at diagnosis. Besides, the overall DNA methylation patterns were accentuated at relapse AML patients, implying the dynamics of DNA methylation during disease progression [93]. Later, a genome-wide DNA methylation study using the enhanced reduced representation of bisulfite sequencing (ERRBS) technique to compare the DNA methylome in a larger cohort of 138 paired AML patient samples at diagnosis and relapse. The authors observed that patients with a higher proportion of loci, particularly at promoters, which underwent methylation changes at diagnosis had a shorter time to relapse when compared to the group with fewer loci. In the comparison of the differentially methylated loci during leukemia progression, three categories including diagnosis-specific, relapse-specific, and shared loci were defined. Interestingly, a significant number of AML patients showed a predominance of diagnosis-specific (41%) or relapse-specific (30%) loci, which is independent of their age, white blood cell count, French-American-British (FAB) classification, and the burden of somatic mutations [21]. This study thus supports the use of DNA methylation signature as a biomarker for AML progression. Although it remains a challenge in the identification of diagnostic epigenetic biomarker, global projects, such as the International Cancer Genome Consortium (ICGC) and the European Community initiative BLUEPRINT Consortium, pave the way towards the longitudinal analysis of the epigenomic changes in AML patients.

### 4.2. DNA Methylation in the Prognosis of AML

Although the molecular mechanism of AML development through the alterations of DNA methylation is not fully understood, several studies have reported a strong correlation between patients’ DNA methylation patterns and their clinical outcomes [18,96,97]. An early report by Deneberg et al. utilizing luminometric methylation assay to examine DNA methylation patterns in 107 well-characterized AML patients, including 77 CN-AML patients [97]. They reported that patients younger than 65 years old with low global DNA methylation but with strong *CDKN2B* methylation had better clinical outcomes, such as higher complete remission rate, overall survival and disease-free survival [97]. Another study reported by Bullinger et al. which applied the matrix-assisted laser desorption/ionization time-of-flight mass spectrometry (MALDI-TOF-MS) to quantitatively measure DNA methylation at 92 genomic regions in 182 AML patient samples. Apart from the identification of novel leukemia subgroups, the quantitative methylation patterns provide a predictive value on patient survival time. Unexpectedly, such a methylation-based prediction model can only be applied to the non-CN-AML cases, but not the CN-AML patients [96]. The authors proposed that the choice of candidate gene regions may not be distinctive in the CN-AML patient group, and therefore a comprehensive genome-wide DNA methylation analysis should be implemented to identify the most predictive epigenetic loci.

Global DNA methylation patterns can be used to assist genetic characterization for further stratification of AML risk groups. For instance, the AML patient group with *CEBPA* mutations was found consisting of two clusters based on the differences in DNA methylation patterns [18]. One cluster with exclusively *CEBPA*-double mutation showed a markedly hypermethylated profile and had a better prognosis than the favorable-risk group. On the other hand, patients with *NPM1* mutations can be sub-divided into four methylation clusters with significant differences in their clinical outcomes. Importantly, the authors also recognized five DNA methylation clusters in CN-AML patients with no morphological or molecular features [18]. This highlights the clinical potential of utilizing DNA methylation for prognosis, especially for the CN-AML group. Besides, it is reported that pediatric AML exhibits distinctive genetic abnormalities to the adult AML [98,99]. A recent study analyzed 284 pediatric AML cases from the Therapeutically Applicable Research to Generate Effective Treatments (TARGET) and TCGA databases and clustered into 30 DNA methylation signatures in association with cytogenetics [100]. Importantly, two of these signatures showed significantly poor event-free survival, suggesting that DNA methylation can also stratify pediatric AML patients. To establish a routine clinical examination of DNA methylation, Luskin et al. applied the expedited microsphere *Hpa*II small fragment enrichment by ligation-mediated PCR (xMELP) assay, which enables a simultaneous assessment of the DNA methylation status at 17 previously identified prognostic loci [101] for calculating a methylation-based risk score (M-score) in a cohort of 166 de novo AML patients who received induction chemotherapy [22]. Strikingly, the reduced multivariable models showed that the M-score has a stronger association with the overall survival and the achievement of complete remission when compared to other prognostic factors, such as cytogenetics, *FLT3-ITD* status, and other genetic lesions. The M-score association was further validated in multiple independent AML cohorts [102], indicating its implication for prognostication.

DNA methylation is classically defined by 5mC; however, 5hmC has been proposed as another clinical biomarker with prognostic value. In fact, deregulated 5hmC level was reported as a predictive marker for prognosis and survival in several types of cancers, such as astrocytoma [103], kidney cancer [104] and esophageal squamous cell carcinoma [105]. For AML, Kroeze et al. performed HPLC-MS/MS to assess the 5hmC levels in 206 younger adult AML patients (≤ age of 60). It was found that the 5hmC levels at diagnosis were significantly different from those of healthy controls or patients at remission. Of note, *MLL* abnormalities and AML1-ETO were enriched in the high 5hmC group, whereas the low 5hmC group exclusively consisted of patients with *TET2* or *IDH1/2* mutations. Patients with high 5hmC levels showed a significantly lower 5-year overall survival rate when compared to the low and intermediate 5hmC groups [106]. However, a recent study showed that 5hmC has limited prognostic value in terms of overall survival, event-free survival and relapse risk in CN-AML patients with *TET2* or *IDH1/2* mutations [107]. Although both *TET2* or *IDH1/2* mutations resulted in a low global 5hmC level, it was previously reported that *TET2* mutations were generally associated with poor prognosis, while the impact of *IDH1/2* mutations on overall survival was less pronounced [108]. This suggests that a low 5hmC level may not be the only parameter that affects the clinical outcomes. Taken together, the 5hmC levels in association with genetic alterations need to be further evaluated for the prognostic stratification of AML patients.

## 5. Targeting DNA Methylation in AML

Allogeneic stem cell transplantation (SCT) is regarded as the only curative treatment for AML [109,110]. However, this treatment is limited by the availability of the human leukocyte antigen (HLA)-matched donors. Some patients, especially those at advanced age, are also not eligible for transplantation [111]. As a result, chemotherapy remains the mainstay treatment for AML patients. The conventional regimen of chemotherapy with anthracyclines/cytarabine includes a high dose for induction and a low dose for consolidation and maintenance. By administration of such regimen, the complete remission rate can be achieved in 70–80% for AML patients younger than 60 years old, but inferior to 40–60% for patients older than 60 years of age. Nevertheless, a high proportion of these patients will relapse without further treatment, such as SCT [112,113]. Considering aberrant DNA methylation plays an important role in AML pathogenesis, therapeutic agents targeting DNA methylation is believed to be a novel therapeutic strategy in AML treatment. In the 1960s, two DNA methyltransferase inhibitors (DNMTi), 5-azacytidine (5-aza-C) and decitabine, were demonstrated to have an anti-leukemic activity in leukemia mouse models [114,115]. With the extensive clinical trials of DNMTi which provide a survival advantage, especially for elderly and SCT ineligible AML patients [116], the European Medicines Agency has approved the AML treatment with 5-aza-C and decitabine in 2008 and 2012, respectively, to patients who are ineligible for induction chemotherapy and/or SCT.

The anti-leukemic effects of DNMTi were mediated through promoting wild-type DNMT degradation [117,118], causing DNA demethylation [119], and inducing cytotoxicity [120]. Generally, a high dose of DNMTi exerts cytotoxic effects on AML cells, while a low dose induces DNA hypomethylation. It was reported that a low dosage of decitabine induced DNA demethylation at the promoters of tumor suppressor genes, such as *CDKN2B* (*p15^INK4B^*) and *E-cadherin*, and thereby reactivated their expression in AML cells [121]. A systematic review by a pooled analysis of five randomized clinical trials of AML patients showed that the DNMTi-treated group has a significantly better overall survival and completed/partial remission than the group with conventional care regimen [122]. Even though DNMTi demonstrates significant improvement in AML treatment, it is noticed that more than half of the AML patients developed resistance and suffered from refractory or rapid relapses [123]. To improve the efficacy of DNMTi treatment, several studies have investigated whether the pre-treatment methylation status correlates with the clinical responses. While several studies reported that a low level of *CDKN2B* promoter methylation before treatment predicted a good clinical response to DNMTi, other studies observed no such correlation [124,125,126,127]. Instead, some studies showed that demethylation of *CDKN2B* after treatment with decitabine, rather than the pre-treatment *CDKN2B* promoter methylation level, was positively correlated with the therapeutic response [128,129,130]. These inconsistent findings of the DNMTi responses associated with the *CDKN2B* promoter methylation could be attributed to the variations in patient cohorts, different methodologies in the quantification of DNA methylation, or co-administration of other epigenetic agents. Apart from the candidate gene methylation, the global DNA methylation level has been considered valuable for the evaluation of the clinical responses to DNMTi. By comparing the global DNA methylation level in 16 AML patients before and after the decitabine treatment, a study reported that the responsive group showed a relatively higher baseline global methylation level and displayed a markedly drop in methylation during the treatment [131]. However, a larger patient cohort is needed to confirm its predictive value in DNMTi response.

DNMTi can be administrated as a sole therapeutic agent; however, it has been proposed that combination with other chemotherapeutic drugs could be more effective in the anti-leukemic therapy. Of note, aberrant epigenetic modifications associated with histone proteins are often observed in leukemia cells. Therefore, a combination of DNMTi (e.g., decitabine or azacitidine) and histone deacetylase inhibitor (HDACi) (e.g., vorinostat or entinostat) showed synergistic anti-leukemic effects in AML and MDS patients [124,132]. In *MLL*-rearranged leukemia, several MLL fusion proteins interact with DOT1L, which is a histone H3 lysine 79 (H3K79) methyltransferase, and induce transcription of target genes, e.g., *HOXA9* and *MEIS1* [133,134,135]. It was reported that 5-aza-C and DOT1L inhibitor EPZ-5676 synergistically inhibited the proliferation of leukemia cells with *MLL* abnormalities [136]. Besides, *FLT3* activating mutations occur in 35% of AML patients. Targeting FLT3 mutants by tyrosine kinase inhibitors (TKIs), such as lestaurtinib (CEP-701) and midostaurin (PKC412), is thus an attractive therapeutic strategy for AML patients. However, most of the AML patients treated with TKIs failed to achieve a durable response, despite the fact that these drugs displayed strong effectiveness in the cell culture system. In fact, a significant proportion of AML patients developed acquired resistance to TKIs over time [137,138]. Interestingly, a study showed that the TKI resistance can be partly associated with the DNA hypermethylation at *SHP-1*, which is a negative regulator of JAK-STAT pathway [139]. This finding leads to the hypothesis that combining DNMTi with TKIs might increase the clinical efficacy in AML treatment. It is indeed supported by both in vitro and in vivo leukemia models showing that the 5-aza-C treatment increased the sensitivity to TKIs [139,140]. Several clinical trials of TKIs in combination with DNMTi (ClinicalTrial.gov identifier: NCT01202877, NCT02752035) have also been conducted to examine their clinical outcomes [141]. Preliminary data from the combined treatment showed high response rates and increased overall survival in AML patients with *FLT3* mutation [142], which warrant further trials with larger patient cohorts.

## 6. Perspectives

DNA methylation holds promising potential in clinical applications, such as early diagnosis, prognostic prediction, and therapeutic decision-making (Figure 2). Even though many research studies have established different methodologies for the assessment of DNA methylation in AML patients, only a few epigenetic biomarkers have been discovered with clinical values. The slow progress in the discovery of DNA methylation biomarker can be caused by, first, a lack of a standardized method for the detection of DNA methylation between studies; second, most of the current methods do not distinguish the heterogeneous DNA methylation between alleles [143]; third, a lack of reproducibility in different patient cohorts. Therefore, a robust way of quantitatively detecting DNA methylation for the routine clinical practice should be warranted in the early diagnosis and guidance of personalized care for AML patients.

For the therapeutic aspect, DNMTi mediates global demethylation regardless of the genomic loci, which may trigger undesirable cellular responses and cytotoxicity. A recent development of the CRISPR with deactivated Cas9 (CRISPR-dCas9) system has been utilized in the precise modulation of the epigenetic status at specific loci. Through the fusion of dCas9 with the catalytic domain of DNMT3A, Vojta et al. demonstrated that the fusion protein can specifically introduce DNA methylation at the promoters of *BACH2* and *IL6ST*, which are involved in autoimmune disease [144]. Another study also reported the use of an improved CRISPR-dCas9-Sun Tag-DNMT3A system to target DNA methylation at the *HOX5A* locus for gene silencing [145]. To remove DNA methylation, Morita et al. also employed the CRISPR-dCas9-Sun Tag system to guide the catalytic domain of TET1 to the target regions. This system could attain a demethylation efficiency more than 50% in 7 out of 9 genomic loci [146]. These studies illustrate the feasibility of site-specific epigenome targeting with a minimal impact on the global DNA methylome, which would be one of the major directions in the epigenetic therapy to AML.

## Figures and Tables

**Figure 1 ijms-20-04576-f001:**
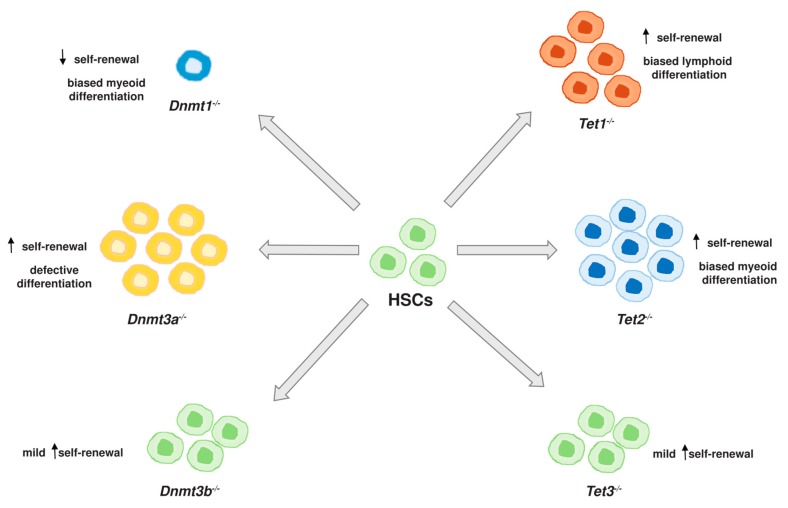
Schematic overview of the impacts of DNA methylation enzymes in hematopoietic stem cells (HSC)s.

**Figure 2 ijms-20-04576-f002:**
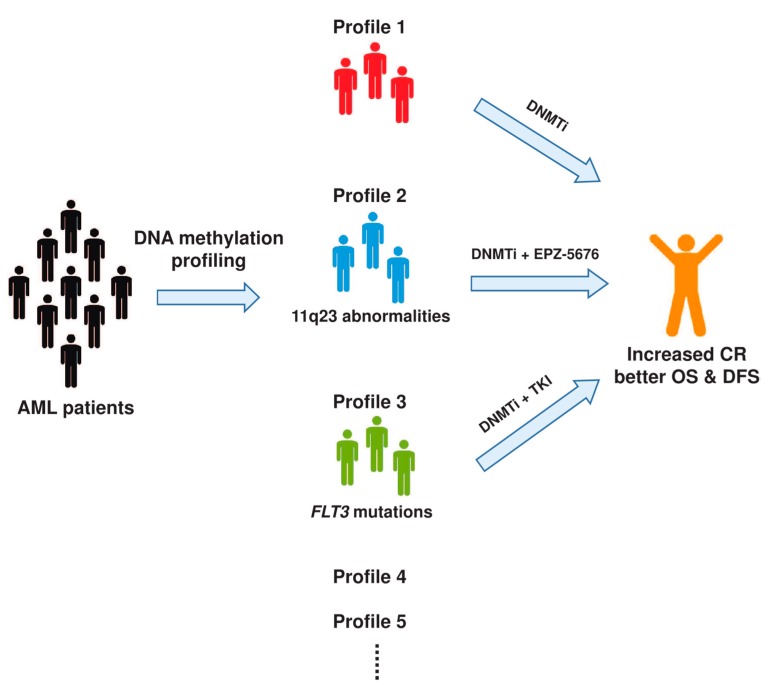
Schematic overview of the clinical implications of DNA methylation in acute myeloid leukemia (AML) stratification and treatment. CR, complete remission; OS, overall survival; DFS, disease-free survival.

**Table 1 ijms-20-04576-t001:** Clinical implications of DNA methylation in acute myeloid leukemia (AML).

Gene(s)	Sample Size	Sample Type	Molecular Method(s)	Clinical Relevance	Reference
*C1R*	194 AML (from TCGA) for analysis; 146 AML for validation	BM	Pyrosequencing	*C1R* hypermethylation is correlated with better OS	[83]
*CEBPA*	193 AML	BM	Bisulfite MassARRAY analysis	*CEBPA* hypermethylation is correlated with better OS	[84]
*DNMT3A*	194 AML (from TCGA)	BM	Human Methylation 450K BeadChip	*DNMT3A* hypermethylation (internal promoter) is correlated with lower EFS and OS	[85]
*GATA4*	105 AML	BM	MSP, BSP	*GATA4* promoter methylation is correlated with MRD and poor OS	[86]
*GPX3*	181 de novo AML	BM	Real-time quantitative MSP	*GPX3* hypermethylation is correlated with poor OS in non-APL AML	[87]
*ITGBL1*	131 AML	BM	Real-time quantitative MSP, BSP	*ITGBL1* hypermethylation is correlated with lower CR rate, poor DFS and OS.	[88]
*MEG3*	45 AML	PB	Combined bisulfite restriction analysis	*MEG3* hypermethylation is correlated with better OS	[89]
*TERT*	33 AML & 10 AML/MDS	PB/BM	Pyrosequencing	*TERT* hypermethylation is correlated with poor OS	[90]
PcG targets (*CDKN2A*, *CDH1*, *HIC1* and *CDKN2B*)	118 CN-AML	BM	Human Methylation 27 BeadChip, pyrosequencing	Methylation of PcG targets is correlated with better EFS and OS	[91]
*CD34*, *RHOC*, *SCRN1*, *F2RL1*, *FAM92A1*, *MIR155HG*, *VWA8*	134 CN-AML for analysis; 355 CN-AML for validation	BM	MethylCap-Seq	Hypermethylation of the selected genes is correlated with better OS	[92]
*NOR1, CDH13, p15, NPM2, OLIG2, PGR, HIN1, and SLC26A4*	30 paired AML (diagnosis and relapse)	BM/PB	Pyrosequencing	All the selected genes have increased methylation at relapse	[93]
*BARD1*, *BCL9L*, *CLEC11A*, *DEFB1*, *FOXD2*, *IGF1*, *IL18*, *ITIH1*, *LSP1*, *P2RX6*, *RNASE, TUBGCP2*	21 de novo AML for analysis; 169 from TCGA for validation	BM	MethylCap-Seq	High M-value is correlated with lower CR, increased hazard for DFS, and poor OS	[94]
*BTBD3, CXCR5, E2F1, FAM110A, FAM30A, GALNT5, KIAA1305, LCK, LMCD1, PRMT7, SLC7A6OS, SMG6, SRR, USP50, VWF, ZFP161*	344 AML	BM	HELP	Distinct DNA methylation patterns defines new AML subtypes. Methylation of the selected genes is predictive of OS	[18]
*CCDC85C, CHL1, ELAVL2, FAM115A, FAM196A, GRP146, GPR6, HELZ2, ID4, IL2RA, KCNG3, LOC254559, LOC284801, NPAS2, PCDHAC2, PROB1, SHISA6, SLC18A3, SOCS2, TRIM67, ZFP42*	138 paired AML (diagnosis and relapse)	BM	ERRBS	Higher number of loci with differential methylation at diagnosis is correlated with shorter time to relapse	[21]

BM, bone marrow; BSP, bisulfite sequencing PCR; CR, complete remission; DFS, disease-free survival; EFS, event-free survival; ERRBS, enhanced reduced representation of bisulfite sequencing; HELP, *Hpa*II tiny fragment enrichment by ligation-mediated PCR; MRD, minimal residual disease; MSP, methylation-specific PCR; OS, overall survival; PB, peripheral blood.

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
