# Peer review of "Aberrant DNA Methylation in Acute Myeloid Leukemia and Its Clinical Implications"

_ijms, 2019, doi:10.3390/ijms20184576_

Round 1

Reviewer 1 Report

In the present manuscript, Yang et al summarize the advances of the field of DNA methylation in the diagnosis and prognosis of Acute Myeloid Leukemia (AML). The work starts by an introduction of normal and aberrant DNA methylation in AML. Then the authors discuss different AML subgroups with regards to specific patterns of DNA methylation. This is followed by a discussion of the present literature of clinical application of DNA methylation analysis and its translation to diagnosis and prognosis in AML. In the last chapter, they discuss the currently available treatment options that directly interfere with DNA methylation.

The manuscript provides a comprehensive overview of the field and the current clinical developments of DNA methylation analysis. I have a few suggestions:

1 – It would be very helpful to include a figure that summarizes the most important aspects of DNA methylation in hematopoiesis and AML

2 –While roughly 50% of the manuscript’s content describe normal and altered DNA methylation in AML without a clinical connection, the selection of a different title could better illustrate the content of this work

3 – Chapter 2 (DNA Methylation Enzymes in Hematopoiesis), lines 104 ff: Can the authors discuss the roles of TET1/2/3 enzymes and any studies that have analyzed redundant and specific effects among the members of this protein family?

4 – Chapter 3 (Aberrant DNA Methylation in AML): It is not clear why the authors highlight the AML subgroups APL (with PML-RARa) and AML with MLL-Gene Rearrangements before discussing AMLA-related DNTM3A and TET2 mutations. For the sake of clarity, I suggest to remove chapters 3.1. and 3.2. and focus on chapters 3.3. (there is a typo in the header!) and 3.4.

5 – Chapter 3.3.: can the authors discuss the mouse model work that has shown the cooperativity of DNMT3 mutations with other AML oncogenes?

Minor:
- Line 33: rephrase: "complex" instead of "complicated"
- Meis1 is frequently "Mesi1" throughout the manuscript. Please correct this.
- Lines 265-267: Rephrase and be more specific. Are the 8 genes methylated in all 21 cell lines and 30 patients, or just in some of them?

- Lines 350-353: Say that these inhibitors are only approved for patients who are unfit/ineligible for induction chemotherapy and/or HSCT
- Line 400: Please write one sentence explaining the results from these clinical trials.

Author Response

Point 1: It would be very helpful to include a figure that summarizes the most important aspects of DNA methylation in hematopoiesis and AML

Response 1: We thank Reviewer for the comment. We have included Figure 1 to summarize the impact of different DNA methylation enzymes on HSC biology.

Point 2: While roughly 50% of the manuscript’s content describe normal and altered DNA methylation in AML without a clinical connection, the selection of a different title could better illustrate the content of this work

Response 2: We now change the title to “Aberrant DNA methylation in Acute Myeloid Leukemia and its Clinical Implications” for better illustration of the content.

Point 3: Chapter 2 (DNA Methylation Enzymes in Hematopoiesis), lines 104 ff: Can the authors discuss the roles of TET1/2/3 enzymes and any studies that have analyzed redundant and specific effects among the members of this protein family?

Response 3: We have modified the section about TET enzymes in Chapter 2 by including more information about their specific functions and redundant effects in hematopoietic development (line 105-114, 130-135).

Point 4: Chapter 3 (Aberrant DNA Methylation in AML): It is not clear why the authors highlight the AML subgroups APL (with PML-RARa) and AML with MLL-Gene Rearrangements before discussing AML-related DNTM3A and TET2 mutations. For the sake of clarity, I suggest to remove chapters 3.1. and 3.2. and focus on chapters 3.3. (there is a typo in the header!) and 3.4.

Response 4: We understand Reviewer’s concern. Instead of focusing on the AML with single gene mutation, we would like to include the AML (or APL) with fusion gene product (owing to chromosomal translocation) for comparison. Therefore, we highlight the leukemia with PML-RARa and MLL fusion proteins in Chapter 3.1 and 3.2. We select these two types of fusion proteins for discussion because of their differences in the dependence of DNA methylation for their maintenance. While aberrant DNA methylation is crucial to MLL-rearranged AML, it is not in the case of PML-RARa APL. We hope this could provide a better (and fair) overview of the impact of DNA methylation in leukemia to the readers.

For clarification, we modified the last sentence (line 149-150) in Chapter 3 as follow:

“we discuss the role of DNA methylation in the pathogenesis of AML with various genetic abnormalities generated from chromosomal translocations or single gene mutations.”

Point 5: Chapter 3.3.: can the authors discuss the mouse model work that has shown the cooperativity of DNMT3 mutations with other AML oncogenes?

Response 5: We have included more information about the knock-in mouse model with Dnmt3amut/Flt3ITD/Npm1c mutations for the discussion of the cooperation of DNMT3A mutation with other genetic alterations (line 217-226).

Point 6: Line 33: rephrase: "complex" instead of "complicated"

Response 6: We change the word to “complex”.

Point 7: Meis1 is frequently "Mesi1" throughout the manuscript. Please correct this.

Response 7: We went through the manuscript and corrected all the spelling of Meis1.

Point 8: Lines 265-267: Rephrase and be more specific. Are the 8 genes methylated in all 21 cell lines and 30 patients, or just in some of them?

Response 8: We rephrase the sentence to “Another study used bisulfite pyrosequencing to screen for methylated CpG islands in 21 leukemic cell lines and 30 primary patient samples had identified 8 genes (NOR1, CDH13, CDKN2B, NPM2, OLIG2, PGR, HIN1, and SLC26A4) that were frequently methylated.”

Point 9: Lines 350-353: Say that these inhibitors are only approved for patients who are unfit/ineligible for induction chemotherapy and/or HSCT

Response 9: We rephrase the sentence to “the European Medicines Agency has approved the AML treatment with 5-aza-C and decitabine in 2008 and 2012, respectively, to patients who are ineligible for induction chemotherapy and/or SCT.

Point 10: Line 400: Please write one sentence explaining the results from these clinical trials.

Response 10: We add a sentence to conclude the preliminary results from these clinical trial: “Preliminary data from the combined treatment showed good response rates and increased overall survival in AML patients with FLT3 mutation, which warrant further trials with larger patient cohorts.”

Reviewer 2 Report

This review summarized the clinical implications of DNA methylation and the current therapeutic strategies of targeting DNA methylation in AML therapy by quoting abundant references.

Although the manuscript is well written in each section, several points should be amended or clarified for the benefit of the readers.

Minor considerations:

1.Recently, several articles of DNA methylation in pediatric AML have been reported. Authors should touch in this topics and compare the differences in the DNA methylation of pediatric compared to adult AML.

2.Authors should show an intelligible figure in the section of targeting DNA methylation in AML.

Author Response

Point 1: Recently, several articles of DNA methylation in pediatric AML have been reported. Authors should touch in this topics and compare the differences in the DNA methylation of pediatric compared to adult AML.

Response 1: We thank Reviewer for the comment. We also notice that pediatric AML shows genetic/epigenetic features which are different from the adult AML. However, the DNA methylation patterns in pediatric AML is not well characterized and we do not aware of any studies that performed systematic comparison between adult and pediatric DNA methylation patterns. Therefore, we choose to focus on the DNA methylation in the adult AML cases. Nevertheless, we have included a brief discussion about pediatric AML DNA methylation signature in Chapter 4.2 (line 338 to 344) to emphasize the prognostic value of DNA methylation in both adult and pediatric AML.

Point 2: Authors should show an intelligible figure in the section of targeting DNA methylation in AML.

Response 2: We thank Reviewer for the comment. We have included Figure 2 to summarize the clinical implications of DNA methylation in AML stratification and treatment.

Reviewer 3 Report

Good comprehensive review. Please add recent 2019 references to have the latest update, (as the paper is seems to be written for late 2018 based on title) In addition, all human gene names need to be italicized)

Author Response

Point 1: Good comprehensive review. Please add recent 2019 references to have the latest update, (as the paper is seems to be written for late 2018 based on title) In addition, all human gene names need to be italicized)

Response 1: We thank Reviewer for the comment. We have corrected human gene names with italic format and included new references published in 2019.